# Simulated Microgravity-Induced Changes in SUMOylation and Protein Expression in *Saccharomyces cerevisiae*

**DOI:** 10.3390/ijms27010042

**Published:** 2025-12-19

**Authors:** Jeremy A. Sabo, Steven D. Hartson

**Affiliations:** 1Biological Preprocessing, Idaho National Laboratory, Idaho Falls, ID 83402, USA; 2Department of Biochemistry and Molecular Biology, Oklahoma State University, Stillwater, OK 74078, USA; hartson.steve@gmail.com

**Keywords:** SUMO, quantitative proteomics, protein modification, SUMOylation, microgravity, cytoskeleton, *Saccharomyces cerevisiae*

## Abstract

Microgravity during space travel induces significant regulatory changes in the body, posing health risks for astronauts, including alterations in cell morphology and cytoskeletal integrity. The Small Ubiquitin-like Modifier (SUMO) is crucial for cellular adaptation, regulating DNA repair, cytoskeletal dynamics, cell division, and protein turnover—all processes affected by microgravity. To determine the extent to which SUMO mediates the cellular response to microgravity stress, *Saccharomyces cerevisiae* cells were cultured under normal gravity and simulated microgravity (SMG) in rotating wall vessels. After 12 h of culture, we investigated changes in SUMO modified proteins and protein expression. We identified 347 SUMOylated proteins, 18 of which demonstrated a 50% change in abundance under SMG. Of 3773 proteins identified, protein expression for 34 proteins decreased and 8 increased by over 50% in SMG (*p* < 0.05). Differentially expressed proteins represented changes in cellular processes for DNA repair, cell division, histone modification, and cytoskeleton regulation. These findings underscore the pivotal role of SUMOylation in orchestrating cellular adaptation to the unique stress of microgravity, revealing potential targets for mitigating spaceflight-induced health risks.

## 1. Introduction

Biological systems exhibit profound changes when exposed to altered gravity, as studied both on the International Space Station (ISS) and in simulated microgravity (SMG) environments. Such conditions lead to significant shifts in cellular regulation, contributing to the pathophysiological alterations associated with spaceflight-related health risks. These alterations include compromised immunity [1,2,3,4,5], muscle atrophy [6], bone loss [7,8,9,10,11,12], nervous system dysfunction [13,14], and organ impairment [15,16,17,18,19,20,21], with evidence suggesting accelerated aging [22,23,24,25,26,27]. Thus, a critical focus in space biology is to identify the precise molecular mechanisms driving these detrimental health effects.

Over the past three decades, studies employing epigenetic, genomic, transcriptomic, metabolomic, and proteomic approaches have been conducted on biological systems exposed to space microgravity and SMG conditions. As of 2024, a comprehensive repository of these investigations has been collected by the Space Omics and Medical Atlas (SOMA) and the international astronaut biobank [28]. Collectively, these investigations have identified responses in several cellular functions, including (i) cytoskeleton dynamics, (ii) mitochondrial function, (iii) DNA mobility and repair, (iv) metabolic reprogramming, and (v) protein turnover [4,5,28,29,30,31,32].

Po et al., 2019 [33] and Gershovich et al., 2012 [9] proposed a model that elucidates microgravity’s impact on cytoskeletal architecture. According to this model, the process begins with mechanosensing at cell membranes, which triggers signal transmission through the cytoskeleton via protein interactions and post-translational modifications. These signals then propagate to the nucleus, influencing nucleoskeleton modifications and thereby altering chromatin dynamics and gene expression. The ultimate outcome of these changes includes modifications in gene expression, epigenetic response factors, and cellular morphology [34,35,36]. However, a disordered cytoskeleton is not a direct conduit for signal transmission as many cellular processes depend on the cytoskeleton for proper functioning [37,38].

The Small Ubiquitin-like Modifier (SUMO) plays a crucial role in regulating a diverse array of proteins involved in key cellular processes, including cytoskeletal structure maintenance [39,40,41], DNA repair [42,43,44], cell cycle progression [45], nuclear transport [46,47], ribosome biogenesis [48], protein turnover [49], and mitochondrial proteins [31,50]—all processes affected by microgravity. SUMO regulates these processes primarily through covalent attachment to target proteins. SUMOylation covalently modifies a protein in several ways: once on a protein (monoSUMOylation), multiple times on the same protein (multi-monoSUMOylation), in a chain on a single protein (polySUMOylation), or on multiple proteins within a complex where non-covalent SUMO interactions further facilitate complex formation (multiSUMOylation) [51,52]. These different forms of SUMOylation extend their influence through non-covalent interactions with adjacent proteins via three distinct noncovalent SUMO binding motifs [53,54]. The ability of SUMO to bind multiple proteins at once can lead to group modifications and protein clustering, resulting in phase separation [49,55]. Furthermore, SUMO cross-talks with other post-translational modifiers, including ubiquitin, phosphorylation, methylation, and acetylation [56,57,58,59]. This versatility results in the formation of SUMO networks that regulate thousands of proteins and various cellular processes.

SUMOylation can alter a protein’s activity, localization, phase separation, or degradation. After SUMO modifies a protein, it is cleaved off by a SUMO-specific protease, generating a free SUMO that can then bind to another protein during cellular processes. The transient nature of SUMO modifications means that SUMO binds to a protein, modifies its function, and is then cleaved off, allowing this process to occur on different proteins at different times during a given cellular process. This dynamic process allows SUMO to participate in multiple steps within a given cellular process and complicates the identification of SUMO’s response to stress.

Previous studies culturing *Saccharomyces cerevisiae* in microgravity have shown that changes in cellular functions mirror those observed in human samples and cell cultures exposed to spaceflight [60,61,62]. However, these phenotypes and changes in gene expression do not fully explain how microgravity drives cytoskeleton dysregulation at its source: protein interactions and post-translational modifications. Gene expression and -omics markers do not always correlate with protein expression or regulation [30,63], highlighting the need to study processes downstream of gene expression. *S*. *cerevisiae* is a suitable model organism as its genome and proteome are well characterized. Additionally, it can reach high cell densities, enabling the capture and quantification of transient cellular processes such as SUMOylation.

To bridge the gap between microgravity and protein activity, we surveyed protein expression and SUMOylation changes in *S*. *cerevisiae* using one-liter rotating wall vessels (RWV). The RWV, designed by NASA, generates a low fluid-shear environment with laminar flow, simulating microgravity by randomized orientation to the gravity vector and continuous free fall [64,65,66]. It is popular for growing spherical cell aggregates due to the equally distributed gravity vector around the tumbling cell [64,66,67,68]. RWVs come in various types, each offering different growth conditions and gas exchange options, fitting inside an incubator and autoclave, making it practical for modulating a microgravity environment [4,69,70,71]. Our NASA-designed RWV was the first of its kind due to its larger one-liter volume capacity.

In this study, we quantified protein expression and SUMOylation changes in *S*. *cerevisiae* cultured under simulated microgravity (SMG). Our findings highlighted the involvement of cytoskeleton proteins and cytoskeleton-dependent processes, supporting the model that altered gravity environments modulate cytoskeleton dysregulation, leading to changes in cytoskeleton-dependent processes [33,36]. To better detect SUMOylated proteins, we used a modified SUMO resistant to Lys-C digestion, which cannot form polySUMO chains. This approach focuses on monoSUMOylation and multiSUMOylation sites. This strain was a gift from the Aragon lab (Cell cycle group, Imperial College London, London, UK) [72].

## 2. Results

*S*. *cerevisiae* cells were cultured in the RWV as described in the Methods and portrayed in Figure 1. After harvesting the cells, we measured the cell culture mass and compared the masses from vertical and horizontal growth. Cell pellet weights were not significantly different between the vertical and horizontal RWVs (*p* < 0.05, n = 6) (Appendix A). Thus, cell proliferation was equally robust in both vertical and horizontal culture vessels (Figure 1).

### 2.1. Isolation and Characterization of SUMOylation Networks

A modified SUMO was used to isolate SUMOylated proteins from SILAC-labeled whole cell extracts. SUMOylated proteins were digested using Lys-C, concentrated with SDS-PAGE, and gel fractions were digested by trypsin, resulting in peptides containing the SUMO remnant that were quantified by mass spectrometry (Figure 2). Mass spectrometry assays identified 268 SUMOylated peptides (Table A1), representing 253 proteins. Of these 253 proteins, 65 had been previously identified in other mass spectrometry screens from *S*. *cerevisiae* [72,73,74,75,76,77,78,79,80,81,82,83,84,85] (Appendix A).

MultiSUMOylation involves networks of connected proteins, where the amount of SUMOylation depends on the protein network’s function or the stage of the cellular process. To identify SUMOylated networks including the novel SUMOylated proteins, all 253 SUMOylated proteins were assayed using the STRING analysis and filtered for physical protein interactions. This analysis identified clusters of SUMOylated proteins involved in the cytoskeleton-dependent processes of DNA replication, DNA damage repair, RNA polymerase function, cell growth control, ribosomal and mitochondrial function, vacuolar function, and protein transport (Figure 3). Many of these processes are known to be involved in multiSUMOylation; therefore, this data supports the identification of multiSUMOylation across multiple regulatory functions from asynchrounously cultured cells.

To further dissect the biological relevance of these 253 SUMOylated proteins, a Gene Ontology (GO) analysis was performed (Figure 3). GO highlighted cytoskeleton processes such as kinetochore binding, plasma membrane tubulation, endocytosis, and actin bundles. Moreover, GO expanded the multiSUMOylation processes, and added lipid activity and intracellular transport as key cellular processes enriched by these SUMOylated proteins. This analysis recapitulated previous findings regarding SUMOylation involvement and identified 188 new candidate SUMO targets and cellular processes worthy of future investigations into microgravity and protein modifications.

### 2.2. SUMOylation Is Altered Following Growth Under SMG Conditions

To determine the extent to which SUMOylation regulated cellular responses to simulated microgravity (SMG), SILAC labeling was employed to compare SUMOylation in gravity cultures versus SMG cultures. Results showed that of the 253 SUMOylated peptides, 162 peptides were quantified in both the heavy-labeled and light-labeled SILAC culture pairs. Thirty-eight SUMOylated peptides met the significance threshold of *p* < 0.05 for the level of SUMOylation after growth under SMG compared to control conditions (Table 1). This set of 38 SUMOylated peptides includes those exhibiting varying levels of SUMOylation, suggesting that observed changes may reflect shifts in either the expression of the target protein, alterations in the degree of SUMOylation, or a combination of both.

Changes in SUMOylation for all proteins are presented, as there is currently no literature investigating SUMOylation in SMG. To maintain focus on the prominent model that postulates that the cytoskeleton interprets gravational forces, we examined the four cytoskeleton-interacting proteins that exhibited altered SUMOylation levels in response to SMG conditions: Stu1, Kar3, Fin1, and Rvs167. SUMOylated Stu1 displayed a log2 fold change of 2.103, indicating a significant increase in the level of SUMOylated peptides. In contrast, Kar3 and Fin1 exhibited log2 fold changes of −0.473 and −0.134, respectively, indicating decreases in SUMOylation levels under SMG growth conditions. Rvs167 showed a log2 fold change of 0.033, indicating no significant change.

To characterize the cellular responses associated with the increases and decreases in SUMOylation levels, changes were visualized using TheCellMap, a *S*. *cerevisiae*-specific visualization tool [86]. To include the four cytoskeleton proteins, an arbitrary fold change cutoff of 0.134 was chosen. As shown in Figure 4, SUMOylated proteins with increased levels under SMG conditions were involved in processes such as mitosis, tRNA modifications, and ribosome biogenesis, along with three other clusters not directly associated with specific processes. In contrast, SUMOylated proteins with decreased levels were involved in rRNA processing, ribosome biogenesis, mRNA processing, protein folding, nuclear transport, cell wall modification, metabolism, and another unspecified cluster.

### 2.3. Differential Protein Expression of S. Cerevisiae Growth Under SMG Conditions

The changes in SUMO peptides above could have reflected changes in the expression of the target protein, changes in the degree of its SUMOylation, or both. To distinguish between these possibilities, we quantified the global protein expression in SMG vs. control cultures and compared this data to the changes in SUMOylation that were described above. For this, SILAC assays were used to compare proteomes of yeast asynchronously cultured under SMG versus normal gravity (n = 6). These assays identified 3773 proteins and quantified 2877 protein ratios between gravity versus SMG after 12 h (Figure 5). In this set of 2877 proteins, 352 proteins show significant change (*p* < 0.05) with 33 of those proteins by at least 50% (log_2_ fold change ≥0.58) (Table A1). Of these, 28 proteins showed a decreased expression and 5 proteins showed an increased expression relative to the control.

Of particular interest in this small dataset, the SUMO conjugating enzyme UBC9 showed decreased protein expression in response to SMG. Given that UBC9 attaches SUMO to targat proteins, a decrease in UBC9 suggests overall protein SUMOylation would be decreased. Consistent but limited by the small dataset of SUMOylated proteins, protein SUMOylation does lean towards less SUMOylation versus a greater number of SUMOylated proteins (Table 1).

In the GO analysis for our 33 differentially expressed proteins (DEPs) (Table 2), 10 categories were enriched above the background (Table 3). Some categories were involved with the cytoskeleton, cytoskeleton-dependent processes, and post-translational modifications. Notably, post-translational modifications of SUMO, ubiquitin, and acetylation, which bind to a lysine on a target protein and are known to regulate cytoskeleton structure and function, were observed [39,87,88,89,90]. GO enrichments for our DEPs included the nucleotide-excision repair factor 2 complex and the Smc5-Smc6 SUMO ligase complex involved in DNA damage repair.

SMG-induced changes also highlighted key components, such as the mitotic spindle midzone and polar microtubules, which facilitate the growth and stability of microtubules, and the kinesin complex, a microtubule motor protein that transports cargo along microtubules. The Cdc50p-Drs2 complex, involved in cargo membrane transportation, and the ER-plasma membrane contact site, which physically connects these two membrane-bound organelles, were also identified. Additionally, the NuA3b histone acetyltransferase complex and the Cul8-RING ubiquitin ligase complex, both implicated in post-translational modifications similar to SUMO, were enriched in the GO analysis. The PAN complex, functioning as a Poly(A)-specific ribonuclease, was also identified.

### 2.4. Microtubule-Associated Proteins Under SMG Growth Conditions

Using the model that altered gravity conditions disrupt cytoskeleton regulation, and identifying cytoskeleton interacting proteins in our datasets, we cultured a microtubule-associated protein Stu2 in the RWV and compared protein abundance differences. Our lab previously demonstrated that Stu2 toggles between microtubule nucleation and plus-end growth [91], leading to the hypothesis that Stu2 expression would be altered in response to SMG conditions. To assess changes in Stu2 expression, yeast whole cell lysates expressing Stu2-HA on a CEN plasmid showed increased Stu2 in the SMG condition compared to the gravity control (Figure 6A). Anti-HA immunoprecipitations of these whole cell lysates also showed increased Stu2 band intensity under SMG growth conditions compared to gravity control (Figure 6B).

### 2.5. Differential Protein and Gene Expression of S. cerevisaie Cultured Under SMG Growth Conditions

To better understand the relationship between changes in protein expression observed in our SMG cultures, we compared these changes to the mRNA expressions characterized by Sheehan et al. [62] (Figure 7). Both studies employed a RWV experimental design to culture *S*. *cerevisiae* for 12 h (~5 to 6 cellular divisions) at 30 rotations per minute in YPD broth at 30 °C. Sheehan observed 1372 significant changes in gene expression. When comparing our changes in protein expression to the mRNA expression changes reported by Sheehan et al., we found little evidence of correlated protein-mRNA responses. This lack of correlation may reflect well-documented biological differences between mRNA and protein responses [92,93], might have resulted from technical differences between our studies and those of Sheehan et al., or be attributed to a cell-wide dysregulated cytoskeleton.

### 2.6. Connections Between Protein SUMOylation and Differential Protein Expression

We compared changes in protein expression to changes in levels of SUMO peptides from those proteins. Our analysis included 162 SUMOylated peptides and 2877 whole cell extract proteins. Among these, 91 proteins were quantified in both datasets, resulting in a Pearson’s correlation coefficient of r = 0.168, indicating no correlation between the datasets (Figure 8). This lack of correlation indicated that the alterations in SUMO peptides did not reflect simple up- or down-regulation in the expression of the underlying parent protein, but instead implicated potential changes in SUMOylation itself. Specifically, four proteins with statistically significant changes in SUMOylation and protein expression show opposing trends. Specifically, Stu1 showed increased SUMOylation and decreased protein expression. Meanwhile Yfr006W, and uncharacterized peptidase, and FSH3, a serine hydrolase had an increased protein expression and decreased SUMOylation. Pep1, a vacuolar sorting component of the Golgi that transports along the cytoskeleton, had decreased expression and decreased SUMOylation.

## 3. Discussion

This study serves as a probe into SUMOylation under SMG conditions, revealing a profile comprising 253 SUMOylated sites in *S*. *cerevisiae*, with 38 proteins quantified for their changes in differential expression. The use of a SUMO mutant resistant to Lys-C digestion both increases SUMO site detection by mass spectrometry and reduces false positives from ubiquitin which also leaves a glycine di-peptide remnant after trypsinolysis. The tradeoff for increased SUMOylation detection is the prevention of polySUMO chain formation. PolySUMOylation is not essential; however, it facilitates ubiquitin-dependent protein degradation. Given the potential disruption of protein degradation in response to SMG, we listed all SUMO sites that met our *p*-value threshold without filtering by foldchange. The new candidate SUMO sites we identified (Table A1) expand our knowledge of regulatory mechanisms for these proteins. The validity of these SUMO sites is supported by the 65 already identified SUMO sites and their consistency with known overlapping cellular processes.

The modest changes in protein expression, confined to a small number of proteins, are consistent with the phenotypes observed for *S*. *cerevisiae* cultured in SMG [60,62,94,95,96], and supported by the precision of our SILAC assays and their power (n = 6). Importantly, the differential protein expression changes and changes in SUMOylation are associated with the same cellular processes known to be impacted by microgravity, such as cytoskeletal dynamics, DNA repair, and mitochondria function. While the function of these proteins are involved in the observed phenotypes, many of these specific proteins have not been investigated for their contribution to the disordered phenotypes in SMG. Therefore, this dataset provides novel candidates for future studies. Additionally, changes in protein SUMOylation and SUMO regulation via UBC9 demonstrates the involvement of SUMO in response to SMG-induced stress. SUMO is well established for its stress response in normal gravity.

Cytoskeleton organization is essential for effective DNA repair [97,98]. In our data, DNA repair proteins Rad23 and Rad2 exhibited the most significant decrease in expression compared to other proteins (Table 2). While early hypotheses attributed DNA damage in space to radiation exposure, markers of DNA damage repair have also been observed in simulated microgravity environments, radiation-shielded settings, and across various cell types cultured in RWVs [99,100,101,102,103]. Given that DNA mobility, expression, and repair processes are reliant on the cytoskeleton, and considering that cytoskeletal mechanotransduction is disrupted under microgravity conditions, the cytoskeleton emerges as a key candidate for regulating DNA damage responses in these environments [35,98,104]. This highlights the integral role of cytoskeletal dynamics in maintaining genomic stability under altered gravitational conditions.

Proper cytoskeleton organization is essential for both cell function and lifespan. In yeast, bud scars serve as indicators of cell division and failed cellular division, representing cytoskeletal remnants that signal aging [95,105,106,107,108,109]. *S. cerevisiae* has been analyzed using the RWV system to assess replicative lifespan—the total number of divisions a yeast cell can undergo—and chronological lifespan, which measures how long non-dividing yeast cells can survive. Under SMG growth conditions, there was a decrease in replicative lifespan, while chronological lifespan remained unaffected. Interestingly, traditional life-extension strategies, such as the deletion of Sir2—a NAD+-dependent histone deacetylase [110] with a human homolog SIRT1 [111]—further diminished replicative lifespan in SMG conditions [95]. In contrast, caloric restriction was found to extend replicative lifespan under SMG conditions. The reduction in replicative lifespan due to SMG, which is not counteracted by methods effective under normal gravity, supports that alternative mechanisms govern lifespan regulation in SMG.

In human cells, neuronal aging is linked to microtubule disorganization, which may be one pathway through which microgravity accelerates aging [112,113]. Tau is the most extensively studied MAP known for its role in stabilizing and modulating microtubule function [114]. In our study, we identified the MAPs Stu1 (Table 3) and Stu2 (Figure 6) as having altered protein expression under SMG conditions. Specifically, Stu1 exhibited increased SUMOylation (Table 1) and decreased protein expression (Table 3), while Stu2 demonstrated increased protein expression. Stu1 stabilizes microtubules to prevent depolymerization [115] and facilities microtubule cross-linking [116,117]. Stu2 facilitates microtubule nucleation [118] and polarized growth [91]. We suspect that the reduction in Stu1 protein expression, coupled with increased expression of Stu2, contributes to the formation of shorter and disorganized microtubules which is observed in various cell types [119,120,121]. Furthermore, since Stu1 is SUMOylated on the same domain that binds to β-tubulin [122], this suggests that SUMOylation may play a role in mediating Stu1-tubulin interactions, thereby supporting the link between cytoskeletal dynamics, SUMO, and the cellular aging processes.

The architecture of the cytoskeleton is dynamically regulated through interactions between microtubule-associated proteins and individual cytoskeletal components [123]. These interactions can lead to either the formation of higher-order structures or the disassembly of cytoskeletal elements, depending on the cellular context and external stimuli [124]. Protein SUMOylation serves as a mechanisms by which cells toggle cytoskeleton proteins between a state of cytoskeleton growth or disassembly [39,40,87,125,126,127,128,129,130] at the cell membrane [131] and the nucleoskeleton [132], consistent with the changes in SUMOylation we found for nuclear transport, mitosis, ribosome biogenesis, and DNA replication and repair (Figure 4). Given that the cytoskeleton is disorganized under microgravity conditions, the proteins that regulate cytoskeleton function become critical mediators of cellular responses, as their SUMOylation status influences the balance between structural integrity and dynamic adaptability in this altered environment. This regulatory capacity is particularly relevant given that cytoskeletal disorganization is associated with aging and various microgravity-related diseases.

SUMOylation relies on a specific conjugating enzyme, UBC9, and a SUMO-specific protease, which together maintain the steady-state abundance of SUMO proteins necessary for normal cellular function. Previous studies have shown that SUMO levels can increase in response to oxidative stress [133]; other stressors decrease SUMOylation [134,135] or regulate UBC9 directly through oxidation [134] or by SUMOylation of UBC9 [136]. In our data, however, we observed that UBC9 exhibited decreased protein expression under SMG (Table 2), yet we did not detect a corresponding decrease in SUMOylation levels for all identified proteins. This unexpected decrease in UBC9 raises intriguing questions about the regulatory mechanisms of SUMOylation in the context of microgravity. Specifically, while UBC9 expression declined, the maintenance of SUMOylation levels for certain proteins suggests that other factors or compensatory mechanisms may be at play, warranting further investigation into how SUMOylation is modulated under altered gravitational conditions.

Our findings demonstrate that multiple SUMO sites were identified on physically connected proteins (Figure 3), supporting multiSUMOylation cross-talk between proteins. The significance of these events has been established, as multiSUMOylation enhances protein–protein interactions, promotes the formation of functional complexes [137], and regulates cellular signaling pathways [138,139]. Notably, SUMO has been described as a “glue” that stabilizes these interactions [140]. We observed potential multiSUMOylation events for cytoskeleton proteins and cytoskeleton-dependent processes from cell division, DNA replication and repair, ribosome biogenesis and mitochondria function.

SUMOylation in mitochondria is a more recent discovery compared to nuclear and cytosolic SUMO activity [141,142]; however, SUMOylated mitochondrial proteins were identified in our multiSUMOylation probe (Figure 3) and exhibited both increased and decreased SUMOylation in response to SMG (Table 1). Notably, mitochondrial proteins represented the most abundant differentially expressed proteins under SMG conditions (Table 3). Interestingly, while some SUMOylated mitochondrial proteins showed increased levels, others exhibited decreased or unchanged SUMOylation in response to SMG. This observation aligns with the lack of correlation between protein expression and SUMOylation levels depicted in Figure 8. The presence of both increased and decreased mitochondrial proteins, along with the corresponding changes in SUMOylation, reinforces the idea that alterations in protein expression do not directly correlate with SUMO events. Instead, the role of SUMOylation may depend on the specific protein or interactions with adjacent proteins within the complex, further supporting the role of multiSUMOylation.

A mechanistic model proposed by Po et al. (2019) [33] and Gershovich et al. (2012) [9] and reviewed by Wu et al. (2022) [36] provides a framework for understanding the impact of microgravity on cellular processes. This model posits that the initial response to microgravity involves mechanosensing at the cell membrane, followed by signal transmission through the cytoskeleton via protein interactions and post-translational modifications, ultimately influencing gene expression and epigenetic response factors. Our findings support this model by demonstrating differential protein expressions and GO enrichment of mitotic spindles, microtubules, cytoskeleton motor proteins, and post-translational modifications (Table 2 and Table 3). Furthermore, changes in SUMOylation for several cytoskeleton proteins and cytoskeleton-dependent processes were also identified (Figure 3). Lastly, we highlighted the lack of correlation between gene expression and proteins expression (Figure 7), which we purpose is derived from multiple cytoskeleton-dependent processes being regulated at the protein level and influencing gene expression simultaneously.

## 4. Materials and Methods

### 4.1. Strain Table

The strains used in this study (Table 4) were a gift form the Aragon lab [72] or generated in house as described [91].

### 4.2. Yeast Cultivation in the Rotating Wall Vessels 

Yeast cultures were grown in a 1-L rotating wall vessel (RWV) for all experiments (Figure 1). Single colonies were cultured in strain-appropriate media and subsequently inoculated into autoclaved RWVs containing approximately 10 mL of media with an optical density (OD600) of ~2.1, following the Synthecon protocol (Synthecon, Houston, TX, USA). For each experimental setup, two vessels were filled with light media containing ^12^C and ^14^N lysine, while two additional vessels were filled with heavy media containing ^13^C and ^15^N lysine (Sigma-Aldrich, St. Louis, MO, USA). All vessels were incubated at 30 °C and 30 RPM for 12–13 h. CO_2_ bubbles were removed 4 times during the incubation period using the plug and two threaded prongs. There was no aeration for these cultures outside of exposure to atmospheric oxygen during CO_2_ removal. A total of six replicates were cultured for gravity and six replicates for simulated microgravity for each experiment.

### 4.3. Yeast Whole Cell Extract

After 12–13 h of growth, cultures were pelleted by centrifugation (2 min, 600 rcf, 30 °C). The cells were resuspended in 30 mL phosphate buffered saline pH 7.4 (PBS) and then re-pelleted (2 min, 600 rcf, 30 °C). The cell mass was weighed and flash-frozen in liquid nitrogen. The frozen yeast was supplemented with x ml of fresh (same day) PBS-Tween + Protease Inhibitor Cocktail (PIC) (1 mM PMSF, 40 mM 2-Iodoacetamide, 20 mM N-ethylmaleimide, and 1:500 stock Sigma PIC-cat # P8849) and 0.1% tween where x = 0.34% weight of pellet (g). Cell suspensions were cryomilled (Retsch MM400, Haan, Germany) (6 cycles, 2 min, 20 Hz) and submerged in liquid nitrogen between cycles. Cryomilled samples stored at −80 °C.

### 4.4. Whole Cell Extract Clarification

Frozen cryomilled lysates (yRM11575) were resuspended by rocking in ice-cold 0.1 M Tris (pH 7.6) containing 4% SDS and 0.1 M DTT. The samples were then clarified by centrifugation at (14,435 rcf, 30 min, 4 °C) and the supernatant was placed in a fresh tube prior to Bradford protein quantification using a BSA standard curve. The supernatant of each gravity sample was combined with the corresponding simulated microgravity sample at 1 mg:1 mg ratio. (Small aliquots of unmixed samples were used for labeling efficiency described below.) A 5X Laemmli sample buffer (60 mM Tris-HCl pH 6.8, 2% sodium dodecyl sulphate, 10% glycerol, 5% β-mercaptoethanol, 0.01% bromophenol blue) was then added to each sample to generate a 2.5X final sample buffer concentration and the samples were boiled for 5 min and stored at −80 °C before running on 10% SDS-PAGE gels. Gels were stained with Coomassie blue and protein bands were cut into fragments and processed by in-gel digestion using trypsin.

### 4.5. Enrichment and Mass Spectrometry Analysis of SUMOylated Proteins

Cryomilled yeast lysates (yRM11576) were resuspended in cold Buffer A (8 M Urea pH 8, 100 mM NaH_2_PO_4_, 10 mM Tris-HCl, 20 mM NEM, and 20 mM Imidazole) with gentle shaking and were centrifuged (14,435 rcf, 30 min, 4 °C). Supernatant protein concentration determined by Bradford assay. Ni-NTA agarose beads were equilibrated and washed 3 times in Buffer A at a 1:15 bead-to-buffer ratio using centrifugation (500 rcf, 2 min, 4 °C). Then, 4 mg total protein was mixed with 150 µL of equilibrated beads in 1.7 mL Eppendorf tubes, topped off with Buffer A, mixed via inversion (15 RPM, 2 h, 4 °C), centrifuged (500 rcf, 4 °C), and washed 3 times with 1 mL of Buffer A. After the last spin, buffer was removed with a gel loading tip. Proteins were eluted with 100 µL of Buffer A containing 500 mM imidazole. Eluted protein was acetone precipitated (−20 °C for 1 h), centrifuged (18,000 RPM, 15 min, 4 °C), washed three times with 1 mL ice-cold acetone, and concentrated using a speed vac. To digest all proteins except SUMO, protein was digested using Lys-C using the Promega protocol for Lys-C digestion of mixed protein samples (Madison, WI, USA, cat.# VA1170). Protein pellets were resuspended in 8 M urea, 50 mM Tris, pH 8 and quantified using a nanodrop. Methyl methanethiosulfonate replaced 2-iodoacetamide (2IAA) as 2IAA generates the same *m*/*z* shift as a di-glycine modification. Proteins were digested for 18 h at an enzyme to substrate ratio of 1:65. Digestions were stopped with 4% B-mercaptoethanol/2% SDS loading buffer (15 min, room temp) before loading on SDS-PAGE gels. Each biological replicate was resolved on its own 12% SDS-PAGE gel and stained with Coomassie blue. The band corresponding to SUMO and the 10 kDa area above SUMO were in-gel digested with trypsin for mass spec analysis.

### 4.6. HPLC and Mass Spectrometry Identification of Differential Protein Expression

Peptides from the digested samples were dissolved in 0.1% aqueous formic acid and injected onto a 75 cm × 2 cm trap column (ThermoFisher, Waltham, MA, USA, cat # 164705) packed with Acclaim PepMap100 C18, 3 μm beads followed by a 75 cm × 50 cm, PepMap RSLC C18, 2 μm beads analytical column (ThermoFisher, cat # 164942). Peptides were then separated with a 120-min gradient of 3 to 30% acetonitrile/0.1% formic acid and eluted through a stainless-steel emitter for ionization in a Nanospray Flex ion source (Thermo). Peptide ions were analyzed by a high/low mass accuracy approach. Parent ions were measured using the Orbitrap sector of a Fusion mass spectrometer (Thermo), followed by data-dependent quadrupole selection of individual precursors and fragmentation in both the ion trap (CID at 30% energy) and in the HCD cell (30–35% energy). Lastly, fragmented ions were measured again in the Orbitrap sector with a 120,000 resolution. The mass spectrometry proteomics data have been deposited tot eh ProteomeXchange Consortium via the PRIDE [143] partner repository with the dataset identifier PXD070877.

### 4.7. Mass Spectrometry Data Analysis with MaxQuant

MaxQuant [144] version 1.6.1.1 was used to analyze all proteomic data using the *S. cerevisiae* protein database from UniProt (https://www.uniprot.org/, 1 November 2019, 6067 entries). Modified SUMO (Figure 3) was added to the above UniProt database for the covalent SUMOylation analysis. Carbamidomethylation, propionamide, and N-ethylamaleimide of cysteine, oxidation of methionine, acetylation of N-terminal, and EQIGG of lysine were set as variable modifications. Covalent SUMO analysis detection on lysine was di-glycine as a variable modification instead of EQIGG. A multiplicity of 2 was selected to identify the heavy amino acid Lys8 and Arg10. The maximum number of missed cleavages for trypsin digestion was zero. A two-protein rule was used for all analyses except for covalent SUMOylation stub identification, which used a one-peptide rule. The match between runs option was enabled for whole cell extract analysis with a time window of 2 min to match HPLC peaks among the 6 biological replicates and at least 2 technical replicates per bio-rep. The re-quantify option was selected and all other options were left as default. The two injection replicates were merged into one output per biological replicate. For SUMOylated peptides the “GlyGly (K) sites” txt file output from MaxQuant was analyzed using Perseus [145]. FASTA headers, H/L normalized ratios, number of GlyGly (K), GlyGly (K) probabilities, position in peptide, PEP, reverse, and potential contaminates were uploaded into Perseus. Reverse peptides and potential contaminants were removed before performing a 1/x transformation to generate a L/H ratio followed by a log_2_ transform on L/H normalized ratio intensities. Next a 1-sample *t*-test using a *p*-value threshold of 0.05 was performed on the L/H normalized ratios prior to combining the ratios by their median for a singular ratio output. No fold change cutoff was applied to this small dataset to highlight responsive and non-responsive SUMOylation to SMG growth conditions.

### 4.8. Analysis of Whole Cell Extracts Data Using Perseus

The “ProteinGroups.txt” file output from MaxQuant was analyzed using Perseus. Reverse peptides and potential contaminants were removed. Since the simulated microgravity condition used the light (L) media and we want the microgravity conditions on the right of a volcano plot, a 1/x transformation was performed on H/L normalized ratios before performing a log_2_ transform on the L/H normalized ratio intensities. A 1-sample *t*-test using a *p* < 0.05 threshold was performed across the 6 biological replicates using default settings. Proteins from all 6 replicates were merged using their median L/H normalized ratio value to generate a single L/H ratio for each protein. Volcano plots were created using the scatter plot tool with −log_10_
*p*-value *t*-test on the y-axis and normalized log2 L/H ratios on the x-axis. Protein expression criteria were cutoff at Log_2_ fold change (≥0.58), cv (≤0.3), *p* (≤0.05), and q (≤0.05).

### 4.9. SILAC Labeling Efficiency for Heavy Lysine Incorporation

The labeling efficiency for each biological replicate was averaged to derive that heavy lysine accounted for 99.5% ± 0.3 of all lysine detected in the heavy labeled samples and that heavy arginine labeling efficiency was 82.2 ± 7.5 for n = 6. Therefore, only peptides that lacked arginine were used to generate a SILAC ratio for figures in this publication. To generate lysine only peptides, trypsin missed cleavages were set to 0 which reduced our total identified peptides by 5%. By only using peptides that lack arginine, we reduced the total number of proteins in our proteome by 16%.

### 4.10. Growth, HA Immunoprecipitation, and Western Blot of Stu2

Stu2-HA yRM12358 and Stu2-NT yRM12359 were cultured in -LEU synthetic media in the RWVs, collected, flash frozen, and cryomilled as described. The samples were cultured without SILAC heavy media, and vertical and horizontal samples were kept separate. Cryomilled yeast samples were clarified in PBS + 1X PIC at a 1 g/10 mL ratio and centrifuged (14,435 rcf, 30 min, 4 °C). Protin was quantified using the Bradford assay and loaded onto a 10% SDS-PAGE gel for whole cell extract Western blots or enriched by immunoprecipitation. Clarified and quantified yeast extracts were incubated with Santa Cruz anti-HA magnetic beads (Dallas, TX, USA cat # SC-500773) (4 °C, 1 h, 10 RPM) following the manufacturer’s protocol. The beads were collected, washed in cold PBS pH 7.4 containing 0.1% Tween (PBST), transferred to a fresh tube, and washed twice more with cold PBST. HA-tagged proteins were eluted with 2.5X Laemmli sample buffer, boiled for 5 min, and immediately resolved on 10% SDS-PAGE. Proteins were transferred to nitrocellulose membranes and blocked overnight in 0.1% I-block reagent (Applied Biosystems, Waltham, MA, USA) in TBST. Fresh TBST with I-block (ThermoFisher #T2015) was added with primary antibodies the next day. The HA epitope was detected using mouse anti-HA (1:3000 dilution, H7392 lot #C1609, Santa Cruz Biotechnology) and goat anti-mouse-HRP secondary antibody (1:20,000 dilution, 115-036-146, Jackson Immuno Research Labs, West Grove, PA, USA). Actin was detected with Invitrogen mouse MA1-744 (1:2000 dilution) and Jackson goat anti-mouse secondary antibody (1:20,000 dilution). Tubulin was detected with rat anti-α-tubulin MCAB (1:3000 dilution, YSRTMCA78G) and donkey anti-rat Santa Cruz sc-2056 (1:3000 dilution). PGK was detected with mouse anti-Pgk1 (1:2000 dilution, #459250 ThermoFisher Scientific) and goat anti-mouse secondary antibody (1:20,000 dilution, Jackson Immuno Research Labs).

### 4.11. Bioinformatic Analysis

Gene Ontology (GO) was used to generate enrichment outputs and Excel was used to reduce redundancy. The proteome enrichment used only the component of the proteome apparent in our work, versus the whole genome. For the proteome analysis, we identified a limited number of enriched cellular components after correction for multiple comparisons, FDR ≤ 0.1; therefore, we report significantly enriched cellular components *p* ≤ 0.05.

STRING version 11.0 was used to create protein maps. Basic settings of physical subnetwork, confidence network edges, and high confidence for minimum required interaction score were selected with SVG output to ensure the highest resolution.

TheCellMap is a bioinformatic tool created to visualize the interaction network specific for *S*. *cerevisiae* unlike GO or STRING which accommodates multiple organisms. To use TheCellMap 1 November 2019, gene names were imported using default settings. SUMOylated proteins with a log_2_ fold change > 0.134 were added into TheCellMap for increased or decreased SUMOylated proteins. The background color of TheCellMap display was changed to grey prior to saving the image https://thecellmap.org/.

Generative artificial intelligence (GenAI) has not been used in this paper to generate text, data, or graphics, or to assist in study design, data collection, analysis, or interpretation.

## 5. Conclusions

Here, we profiled the proteome and SUMOylated proteins of *Saccharomyces cerevisiae* cultured under SMG growth conditions. Alterations of protein abundance and SUMOylation levels were observed for cytoskeleton proteins and cytoskeleton-dependent processes. Specifically, microtubule associated proteins Stu1 and Stu2 were decreased or increased, respectively, in response to SMG growth conditions. There was no direct correlation between gene expression, protein expression, and protein SUMOylation in response to SMG, reinforcing the model that disrupted cytoskeleton dynamics decouples traditional gene-to-protein regulatory pathways. This indicates that protein SUMOylation functions independently of gene and protein regulation in response to SMG, highlighting a distinct regulatory mechanism. Overall, we provide a quantitative analysis of the proteomic response to SMG, identifying specific proteins and cellular functions affected by altered gravity. Moreover, we unveil previously unpublished SUMOylation sites that contribute new insights into the complex interplay between SUMOylation and cellular dynamics under microgravity conditions. These findings expand our insight into the cellular processes responding to microgravity and introduce the role of SUMOylation.

## Figures and Tables

**Figure 1 ijms-27-00042-f001:**
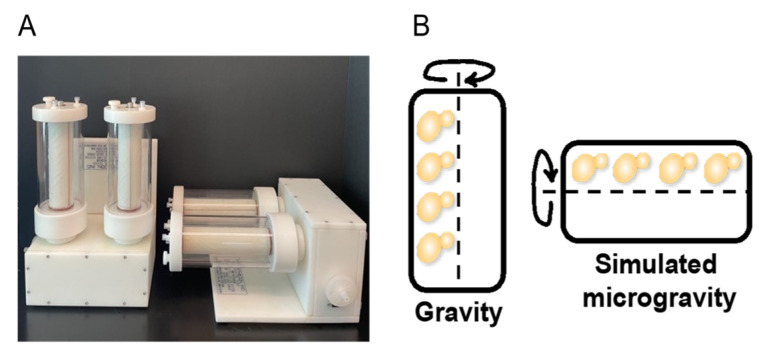
*S*. *cerevisiae* culturing in the 1-L capacity RWV. (**A**) Image of the Synthecon Slow Turning Lateral Vessel (STLV) or commonly referred to as the Rotating Wall Vessel (RWV) (Synthecon, Houston, TX, USA). (**B**) Representation of budding yeast *S. cerevisiae* rotating around the RWV center of axis.

**Figure 2 ijms-27-00042-f002:**
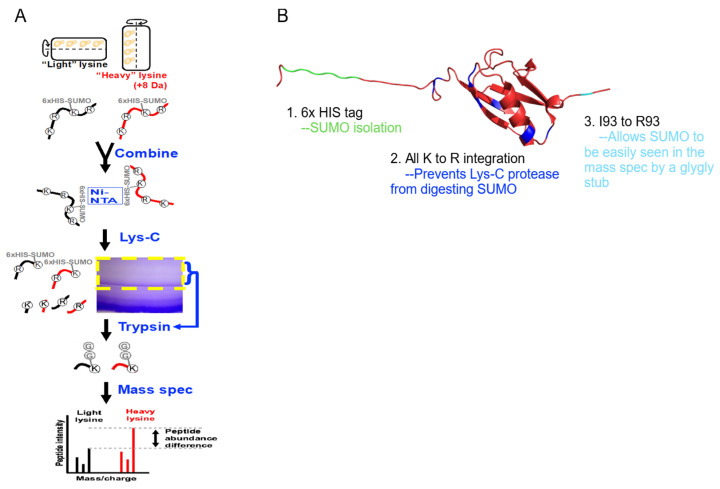
Enrichment of SUMOylated proteins and identifications by mass spectrometry. (**A**) *S*. *cerevisiae* expressing a modified SUMO (SMT3-KallR-I93R) was cultured in SILAC media in the RWV. SUMOylated proteins were isolated, digested with Lys-C, and further enriched by SDS-PAGE. Trypsin digestion reduced SUMO to a di-glycine stub, and the resulting peptides were analyzed by LC-MS/MS. (**B**) PyMol representation of the modified SUMO.

**Figure 3 ijms-27-00042-f003:**
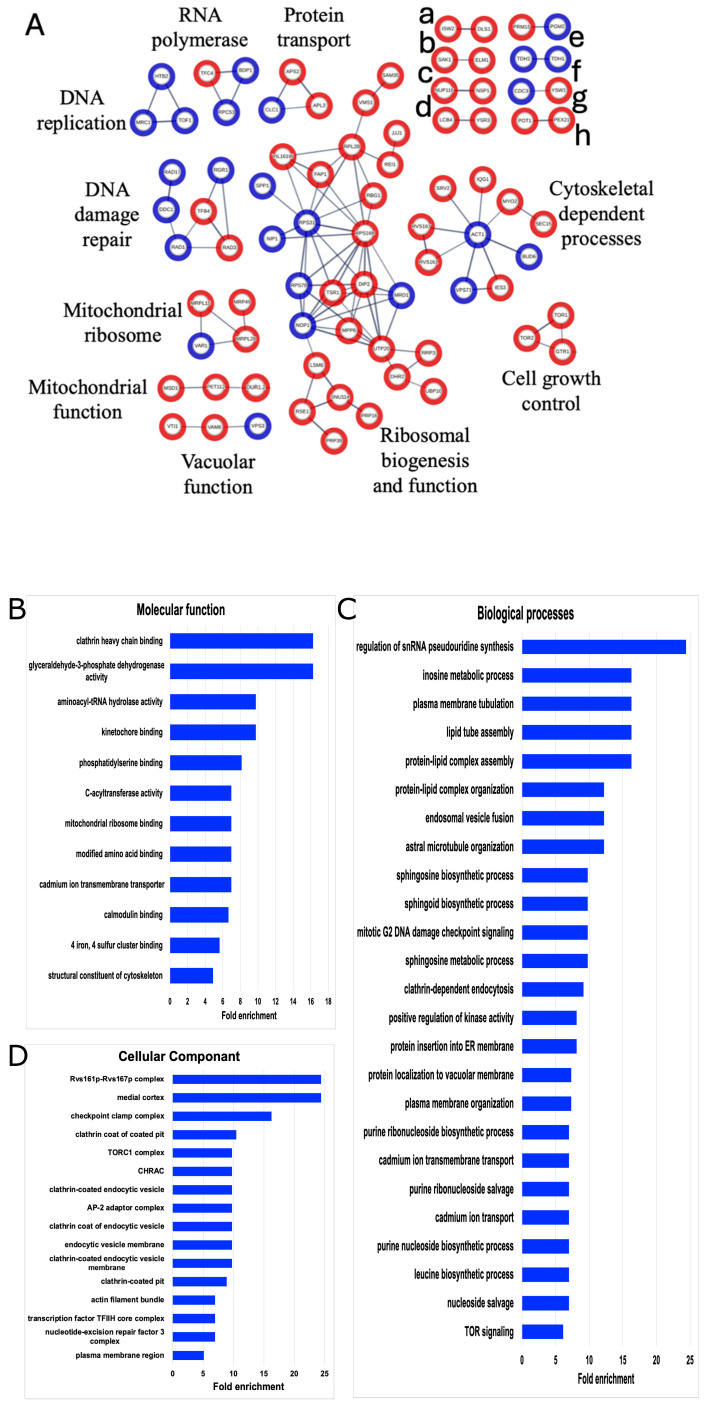
STRING and GO network of SUMOylated proteins. (**A**) The list of 253 SUMOylated proteins was analyzed by STING (version 11.0) and filtered for high confidence (0.700). Only connected nodes with physical subnetwork interactions are displayed. The thicker lines represent greater strength of the data to support the connection. Blue nodes denote SUMOylated proteins seen in both our data and the other published SUMOylation screens. Red nodes are SUMOylated proteins only identified in our dataset. Unlabeled networks include (a) chromatin remodeling, (b) serine/threonine kinases, (c) nucleoporin components, (d) sphingolipid processing, (e) metabolism, (f) GAPDH isozyes, (g) cell growth control, and (h) peroxisomal matrix. Gene Ontology (GO) enrichments for (**B**) molecular functions, (**C**) biological processes, or (**D**) cellular components for the list of 253 identified SUMOylated proteins were constrained by fold change = 5, *p* < 0.05.

**Figure 4 ijms-27-00042-f004:**
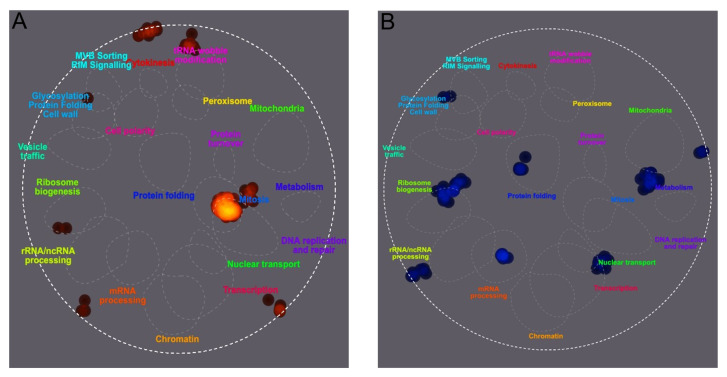
Differential SUMOylation of peptides from *S*. *cerevisiae* cultured under SMG growth conditions versus a gravity control. Yeast was cultured in the vertical or horizontal RWV for 12 h prior to the enrichment of SUMOylated proteins and quantification of SUMOylated peptide ratios using SILAC mass spectrometry. TheCellMap illustrated cellular processes associated with an increase in SUMOylated peptide frequency in red (**A**) and cellular processes associated with a decrease in SUMOylated peptide frequency in blue (**B**).

**Figure 5 ijms-27-00042-f005:**
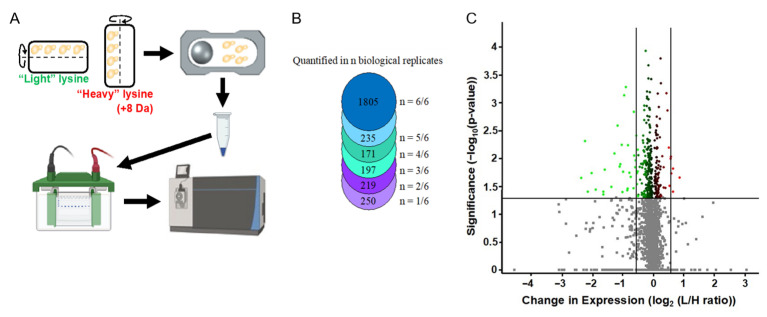
Differential protein expression under SMG conditions. (**A**) Experimental walthrough of protein quantification from *S*. *cerevisiae* cultured in gravity vs. SMG. (**B**) The number of proteins quantified in each biological replicate. (**C**) Volcano plot of protein expression. Green is decreased expression and red is increased expression.

**Figure 6 ijms-27-00042-f006:**
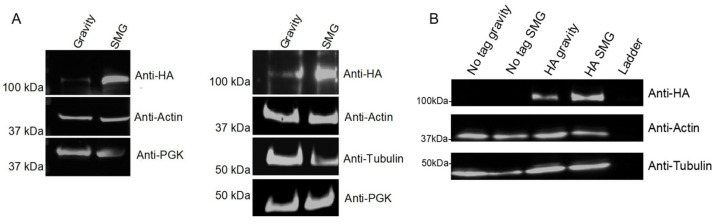
Stu2 protein abundance in gravity vs. SMG. *S*. *cerevisiae* expressing Stu2-HA was cultured for 12 h in gravity or SMG using the RWV. (**A**) Western blots of whole cell extracts (n = 2). (**B**) Whole cell extracts were anti-HA immunoprecipitated followed by Western blotting with anti-HA (n = 1). Actin, tubulin, and PGK served as loading controls.

**Figure 7 ijms-27-00042-f007:**
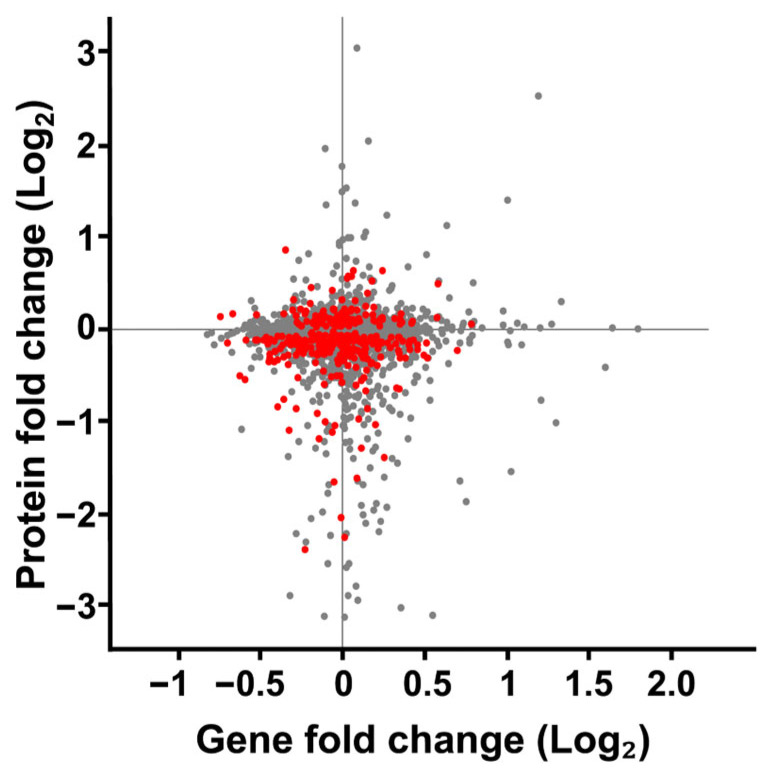
Comparison of fold change in protein expression versus gene expression of *S*. *cerevisiae* cultured in SMG using the RWV. Protein expression changes from *S*. *cerevisiae* cultured in the 1000 mL RWV from this study was compared to gene expression changes *S*. *cerevisiae* cultured in the 25 mL RWV [62]. Each gene product identified in both datasets is represented by a dot on the scatterplot. The protein and gene expression values were subjected to a 1-sample Students *t*-test. Red dots represent proteins that met the *p*-value < 0.05 threshold for change in expression. All gene expression values on the scatterplot met the *p*-value threshold as reported. Grey dots indicate proteins with quantified expression ratios but failed to meet the *p*-value threshold.

**Figure 8 ijms-27-00042-f008:**
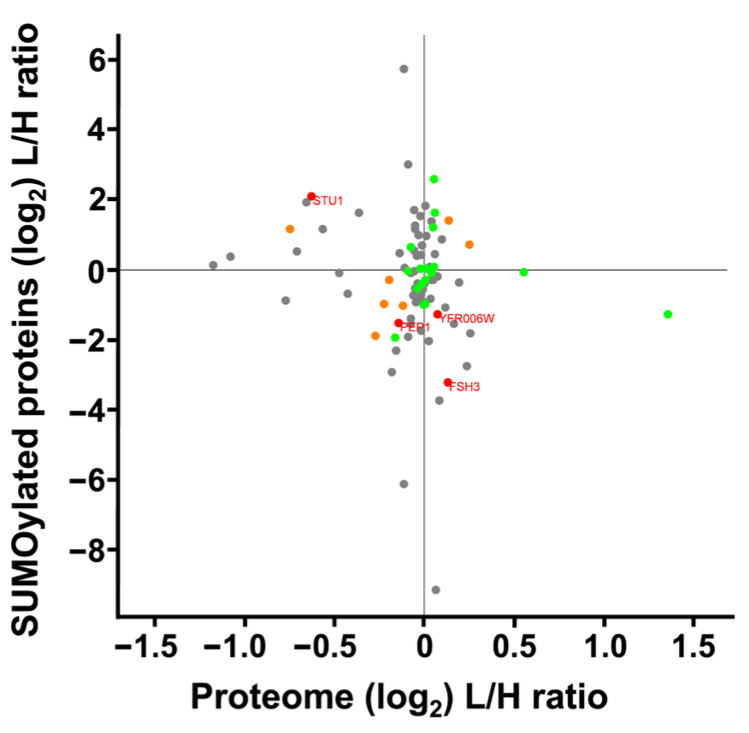
Scatterplot comparing SUMOylation levels vs. differential protein expression of *S*. *cerevisiae* cultured under SMG using the RWV. Proteins whose gene names are displayed showed a statistically significant response to the RWV for both differential protein expression and protein SUMOylation (*p* < 0.05). Green dots indicate statistically significant changes in SUMOylation, orange dots indicate statistically significant changes in differential expression, and grey dots represent proteins identified in both datasets that did not show statistically significant changes.

**Table 1 ijms-27-00042-t001:** The 38 SUMOylated peptides and their differential SUMOylation response under SMG growth conditions. *S. cerevisiae* SILAC cultures expressing a SUMO mutant was cultured in the vertical or horizontal RWV and assayed using SILAC mass spectrometry. Significant (*p* < 0.05) alterations in the expression of each SUMOylated peptide are shown. The location and probability of SUMO lysine conjugation were identified by a di-glycine stub post-trypsinolysis of the engineered SUMO. Probability is displayed as a decimal in parentheses, where 1 is 100% confidence of lysine modification.

Gene Name	GlyGly (K) Localization and Probability	Lysine Position Within Protein	Log_2_ Fold Change	Log_10_ *t*-Test *p*-Value	Protein Description
FSH3	MSEK(0.002)K(0.998)K	5	−3.217	2.050	FSH3_Family of serine hydrolases 3
YKL123W	MK(1)ESLLTLTEK	2	−2.561	2.520	YKM3_putative uncharacterized protein
YGR017W	NNTEPFVSFQFATVDELTNK(1)	37	−1.917	2.639	YG1B_Pyridoxamine-phosphate oxidase-related protein
PEP1	DATGK(O.OO1)CVPDYNLIVLSDVCDK(1)TK(1)	684; 686	−1.507	2.698	VPS10_Vacuolar protein sorting/targeting protein VPS10
TFB4	DEIFQYIPIMNCIFSATK(0.116)MK(0.884)	207	−1.267	3.598	General transcription and DNA repair factor IIH subunit TFB4
YFR006W	SLGYDPICCSGPACGTLHYVK(1)	302	−1.263	3.439	YFH6_ Uncharacterized peptidase
PET191	HNPQECLDNPELNK(0.004)DLPELCIAQMK(0.996)	51	−1.132	4.066	PET191_Mitochondrial protein
ATG4	LQLSEMDPSMLIGILIK(0.074)GEK(0.926)	365	−1.114	2.876	ATG4_Cysteine protease
SAM35	MVSSFSVPMPVK(1)	12	−0.993	4.575	SAM35_sorting assembly machinery
SNU114	K(1)GISTGGFMSNDGPTLEK	974	−0.963	2.352	SNU114_Pre-mRNA-splicing factor
BUR6	ADQVPVTTQLPPIK(1)	15	−0.877	4.460	NCB1_Negative cofactor 2 complex subunit alpha
FCY1	MVTGGMASK(1)WDQK	9	−0.742	1.368	FCY1_ Cytosine deaminase
YGR111W	NKSNEMMTK(1)YPK	65	−0.515	1.320	YG2W_Uncharacterized protein
KAR3	K(1)QFMNEGHEIHLK	172	−0.473	2.052	KAR3_Kinesin-like protein
TDH2	VVDLVEHVAK(1)	331	−0.408	4.224	G3P2_Glyceraldehyde-3-phosphate dehydrogenase 2
KTR2	MQICK(1)YSSSTQA	5	−0.358	4.814	KTR2_Probable mannosyltransferase
HTB2	AVTK(1)YSSSTQA	124	−0.283	2.760	H2B2_Histone
YLR434C	SKAFNICFIK(1)	106	−0.250	4.326	YL434_Putative uncharacterized protein
ECM18	K(1)AGMFMVK(0.975)ELNNLK(0.025)	399; 406	−0.242	1.393	ECM18_Mitochondrial cell membrane protein
YFL021C-A	MLLLFK(1)NGK(1)	6; 9	−0.235	4.017	YF021_Putative uncharacterized membrane protein
HTB2	LAAYNK(0.231)K(0.769)	90	−0.197	2.043	H2B2_Histone
FIN1	IMSPECLK(1)GYVSK	79	−0.134	1.350	FIN1_Cycle-specific filaments between spindle pole bodies in mother and daughter cells
VAC14	MEK(1)SIAK	3	−0.094	1.327	VAC14_Vacuole morphology and inheritance protein 14
VPS3	LTNK(1)CENILK(1)MLLMMK(1)	601	−0.077	4.066	VPS3_Vacuolar protein sorting-associated protein 3
YAP1801	TTYFK(1)LVK	6	−0.070	4.114	AP18A_Clathrin coat assembly protein
TDH1	VVDLIEYVAK(1)	331	−0.038	5.101	G3P1_Glyceraldehyde-3-phosphate
PPS1	ADTVVSDK(1)	669	0.011	5.345	PPS1_Dual specificity protein phosphatase
RVS167	YNGQQGVFPGNYVQLNK(1)	481	0.033	3.047	RV167_Reduced viability upon starvation protein 167
PEX21	PSVCHTSPIEK(1)	12	0.050	1.390	PEX21_Peroxisomal membrane protein
MRPL15	LNLLGAQFLK(1)LQTCIHSLK(1)	116; 125	0.052	2.242	RM15_Mitochondrial ribosomal protein L15
AHP1	MSDLVNK(1)	7	0.097	3.322	AHP1_Peroxiredoxin type-2
GAL4	TVTAEK(0.009)SPICAK(0.991)	459	0.159	2.916	GAL4_Regulatory protein
NUP116	K(0.005)K(0.995)ASLTNAYK(1)	778; 786	0.668	2.181	NUP116_Nucleoporin NUP116/NSP116
RPS7B	MSSVQSK(0.94)ILSQAPSELELQVAK(0.06)	7	1.222	1.487	RS7B_40S ribosomal protein S7-B
APL3	LDPSDEAISNSVTALCSLLTSK(1)	394	1.635	4.204	AP2A_AP-2 complex subunit alpha
STU1	LENNIIYIEEWLK(1)	514	2.103	2.025	STU1_microtubule associated protein
CTR86	NLAAENQEIPQK(1)	109	2.610	4.632	CTR86_Copper transport protein 86
MAL33	CFFDALATESTSGSCTEDSLK(0.455)K(0.545)	305	5.094	2.363	MAL33_ Maltose fermentation regulatory protein
OPI6	MYCEANPPK(1)CTSSNTTLSGHAK	9	5.980	3.320	OPI6_Putative uncharacterized protein

**Table 2 ijms-27-00042-t002:** Differentially expressed proteins in *S*. *cerevisiae* proteome under SMG versus gravity conditions. *S*. *cerevisiae* cultures were compared for differences in protein expression after 12 h when cultured in the vertical versus horizontal RWV. Whole cell extracts of SILAC labeled gravity cultures from the vertical RWV were compared to whole cell extracts of the SMG condition from the horizontal RWV using mass spectrometry. The proteins with log_2_ fold change ≥ 0.58 ratio values that display a statistical significance of *p* ≤ 0.05 for the fold change are listed below. *q* values and protein score listed in Appendix A.

Gene Name	Protein Description	Log_2_ Fold Change (L/H)	Change in Expression
RAD23	RAD23_UV excision repair	−2.38	Decreased
RAD2	RAD2_DNA repair protein	−2.25	
GPN2	GPN2_GPN-loop GTPase2	−2.16	
ART5	ART5_Arrestin-related trafficking adaptor 5	−2.03	
CCM1	CCM1_Mitochondrial group intron splicing factor	−1.90	
RPL21B	RL21B_60S ribosomal protein L21-B	−1.65	
LAM4	LAM4_Membrane-anchored lipid-binding protein	−1.64	
PAN3	PAN3_PAB-dependent poly(A)-specific ribonuclease subunit	−1.60	
FAD1	FAD1_FAD synthase	−1.38	
ZDS1	ZDS1-Regulator of PP2A and Swe1-dependent polarized growth	−1.28	
SAS3	SAS3_Histone acetyltransferase	−1.18	
DSF2	DSF2_Deletion suppressor of mptFive mutation	−1.11	
RTT107	RT107_Regulator of Ty1 transposition protein 107	−1.09	
ARH1	ADRO_Probable NADPH: adrenodoxin oxidoreductase, mitochondrial	−1.04	
KRE29	KRE29_DNA repair protein	−1.03	
MED1	MED1_Mediator of RNA polymerase II transcription subunit 1	−0.99	
ADD37	ADD37_A1-proteinase inhibitor-degradation deficient protein 37	−0.97	
SAM3	SAM3_S-adenylmethionine permease	−0.90	
DRS2	ATC3_Probable phospholipid-transporting ATPase	−0.86	
UBC9	UBC9_SUMO-conjugating enzyme E2	−0.85	
CIK1	CIK1_Spindle pole body-associated protein	−0.83	
CIR1	ETFB_Probable electron transfer flavoprotein subunit beta	−0.79	
LCB2	LCB2_Serine palmitoyltransferase 2	−0.75	
MOT2	NOT4_General negative regulator of transcription subunit 4	−0.66	
YNR066C	YN95_Uncharacterized membrane glycoprotein	−0.64	
STU1	STU1_Microtubule associated protein	−0.63	
UFE1	UFE1_t-SNARE protein retrograde vesicular traffic	−0.60	
SFH1	SFH1_Chromatin structure-remodeling complex subunit	−0.58	
MNE1	MNE1_Mitochondrial mRNA splicing	0.58	Increased
CWC2	CWC2_Pre-mRNA-splicing factor	0.58	
MRPL13	RM1354S_ribosome protein L13, mitochondria	0.64	
YOR352W	TFB6_TFIIH protein complex	0.65	
OAC1	OAC1_Mitochondrial oxaloacetate transport protein	0.86	

**Table 3 ijms-27-00042-t003:** GO analysis of DEPs. The 33 differentially expressed proteins in Table 2 were analyzed for cellular components as described in the Methods.

Cellular Component Complete	Fold Enrichment	*p*-Value
Nucleotide-excision repair factor 2 complex	>100	1.91 × 10^−2^
PAN complexs	51.59	2.85 × 10^−2^
Cdc50p-Drs2p complex	51.59	2.85 × 10^−2^
NuA3b histone acetyltransferase complex	34.39	3.79 × 10^−2^
Kinesin complex	34.39	3.79 × 10^−2^
Mitotic spindle midzone	25.80	4.71 × 10^−2^
Polar microtubule	34.39	3.79 × 10^−2^
Cul8-RING ubiquitin ligase complex	34.39	3.79 × 10^−2^
Endoplasmic reticulum-plasma membrane contact site	25.80	4.71 × 10^−2^
Smc5-Smc6 SUMO ligase complex	25.80	4.71 × 10^−2^

**Table 4 ijms-27-00042-t004:** Strain table for yeast used in this study.

Yeast Strain	Genotype	Source	Experiment/Media
yRM11575/CCG4620	MATa his3∆ leu∆0 met15∆0 ura3∆0 6HisFLAG-smt3Kan MX6	Esteras et al., 2017 [72]	Proteome YPD
yRM11576/CCG9474	MATa trp1-1 ura3-52 his2∆200 leu2-3112 lys2-801 smt3 ∆HIS [6HIS-SMT3-KallR-I93R LEU]	Esteras et al., 2017 [72]	SUMOylation/ YPD
yRM12358	MATa stu2-∆1::HIS3 ura3-52 leu2-3112 his3-∆200 [pRM2119 = pSTU2-STU2-3xHA LEU2+ CEN6 ARSH4 LEU2 AmpR]	Greenlee et al., 2022 [91]	Stu2/ Synthetic deficient -LEU
yRM12359	MATa stu2-∆1::HIS3 ura3-52 leu2-3112 his3-∆200 [pRM6507 = pSTU2-STU2 LEU2+ CEN6 ARSH4 LEU2 AmpR]	Greenlee et al., 2022 [91]	Stu2/ Synthetic deficient -LEU

## Data Availability

The data presented in this study are openly available in the ProteomeXchange consortium via the PRIDE partner repository with the dataset identifier PXD070877.

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
