# Peer review of "Simulated Microgravity-Induced Changes in SUMOylation and Protein Expression in Saccharomyces cerevisiae"

_ijms, 2025, doi:10.3390/ijms27010042_

Round 1
Reviewer 1 Report
Comments and Suggestions for Authors
This study provides a comprehensive investigation into the changes in SUMOylation and protein expression in Saccharomyces cerevisiae induced by simulated microgravity, representing a valuable piece of research. However, I have the following suggestions for improvement:
- Cell pellet weights data should be supplemented. (Page 3, lines 111–112)
- The labels in the STRING network of SUMOylated proteins in Figure 3 should be clearly annotated. (Page 5, line 151)
- To investigate the effect of SMG conditions on stu2 expression, a genomic integration approach would provide more rigorous results. (Page 10, line 261)
- When comparing protein and gene expression levels, it is not appropriate to directly compare the protein expression data from this study with mRNA levels from other studies. Additional data on mRNA expression under the experimental conditions used in this study should be provided. (Page 11, lines 272–274)
Author Response
Thank you for your time in reading and reviewing this manuscript! We have incorporated your comments into the manuscript and feel this draft has been substantially improved.
This study provides a comprehensive investigation into the changes in SUMOylation and protein expression in Saccharomyces cerevisiae induced by simulated microgravity, representing a valuable piece of research. However, I have the following suggestions for improvement:
Comment 1: Cell pellet weights data should be supplemented. (Page 3, lines 111–112)
Response 1: Great note! The cell pellet data has been added to the supplemental section (Fig. S1). Here is the associated figure for cell pellet weight.
Comment 2: The labels in the STRING network of SUMOylated proteins in Figure 3 should be clearly annotated. (Page 5, line 151)
Response 2: The unlabeled groups of SUMOylated proteins have been annotated with letters and referenced in the figure legend (Fig. 3) to avoid clutter and clearly annotate. This is much improved, thank you for the recommendation. (Page 5)
Comment 3: To investigate the effect of SMG conditions on stu2 expression, a genomic integration approach would provide more rigorous results. (Page 10, line 261)
Response 3: Thank you for this note and we agree. We are pursuing collaborators with wild type and genomic integrations for proteins of interest to continue this work.
Comment 4: When comparing protein and gene expression levels, it is not appropriate to directly compare the protein expression data from this study with mRNA levels from other studies. Additional data on mRNA expression under the experimental conditions used in this study should be provided. (Page 11, lines 272–274).
Response 4: Thank you for your attention. We agree that cross study comparisons can result in differences due to experimental conditions, extractions methods, and analysis that are independent of the biological response.
Given the limited data on protein expression from microgravity environments, especially for eukaryotes, we found it valuable to make this comparison. To reduce concern over discrepancies between the studies, we have added more experimental details into the manuscript. These details demonstrate a greater overlap in growth conditions between the two studies. However, we recognize the limitations of this comparison and state, “This lack of correlation may reflect well-documented biological differences between mRNA and protein responses [90, 91], might have resulted from technical differences between our studies and those of Sheehan et al., or be attributed to a cell-wide dysregulated cytoskeleton.” (Page 11, lines 276-284)
Reviewer 2 Report
Comments and Suggestions for Authors
The manuscript under review is an interesting work, which try to show the difference in proteins content and its changes during grow Saccharomyces cerevisiae yeasts in special device with simulation of microgravity versus control cells. That piece of work desired to be published after major revision.
Specific questions:
Q1. Protein scores that were taken into account from the proteomic analysis should be mentioned in the Materials and Methods section: specify the lowest value.
Q2. STRING version and analysis should be mentioned in Materials and Methods.
Q3. Page 5, Figure 3, STRING analysis: protein names are too small to be seen. I suggest to make the separate Figure for STRING analysis.
Q4. Page 5, Figure 3 (C), “Cellular componant” change on “Cellular components”
Q5. Page 5, Figure 3, lines 160 – 162:
To determine the fold change of proteins in proteomic analysis, it is not enough to take into account only p-values ​​(<0.05)! It is also necessary to consider the coefficients of variation (cv ≤ 0.3) and the experimental q-values that are derived from the validation (≤0.05). Excel table on statistical analysis of differentially expressed proteins with UniProt Entries, protein names, initial data on 6 experimental replicates and 2 technical replicates of each growth condition, means, cv-, p-, q-values, fold change, should be submitted as Supplementary Materials.
The data for Figure 3 (A, B, C) should be added to the Supplementary Materials with at least the following information: UniProt Entry, protein name, EC, mol. mass, protein score based on proteomic analysis, gene name, GO.
Q6. Page 9, Table 3. For statistical significance of protein fold change, cv (≤ 0.3) and q (≤ 0.05) values ​​should be assessed in addition to p (≤ 0.05) in the Supplementary materials.
Add the UniProt Entry for each protein to Table 3.
Q7. Page 18, line 594: “saccharomyces cerevisiae” change on “Saccharomyces cerevisiae”
Q8. Ln. 842. Provide page numbers for ref. 119
Author Response
Thank you for your time in reading and reviewing this manuscript. We have incorporated your comments, updated the tables and figures, and added more supplementary tables. Thanks to your suggestions, this draft has been greatly improved.
Specific questions:
Comments 1- Q1. Protein scores that were taken into account from the proteomic analysis should be mentioned in the Materials and Methods section: specify the lowest value.
Response 1- Thank you for this insight. Criteria for proteomic analysis has been added to the Materials and Methods section.
Comments 2- Q2. STRING version and analysis should be mentioned in Materials and Methods.
Response 2- Thank you for noting this lack of meaningful information in Materials and Methods! The version and analysis used for STRING generation is included in the Methods and stated here. The use of high resolution SVG display used in the manuscript allows the viewers to zoom into the figure.
STRING version 11.0 was used to create protein maps. Basic settings of physical subnetwork, confidence network edges, and high confidence for minimum required interaction score were selected with SVG output to ensure the highest resolution. Page 19, line 589.
Comment 3- Q3. Page 5, Figure 3, STRING analysis: protein names are too small to be seen. I suggest to make the separate Figure for STRING analysis.
Response 3- Thank you for this suggestion. Given the high resolution of the STRING analysis, I have increased the size of the figure. Additionally, readers can zoom into the image and obtain clear protein names.
Comment 4- Q4. Page 5, Figure 3 (C), “Cellular componant” change on “Cellular components”
Response 4- Good catch, thank you! This typo has been corrected. As requested by another reviewer, the naming convention of Figure 3 has been updated. The cellular components figure is now Figure 3 (K) on page 5.
Comment 5- Q5. Page 5, Figure 3, lines 160 – 162:
To determine the fold change of proteins in proteomic analysis, it is not enough to take into account only p-values ​​(<0.05)! It is also necessary to consider the coefficients of variation (cv ≤ 0.3) and the experimental q-values that are derived from the validation (≤0.05). Excel table on statistical analysis of differentially expressed proteins with UniProt Entries, protein names, initial data on 6 experimental replicates and 2 technical replicates of each growth condition, means, cv-, p-, q-values, fold change, should be submitted as Supplementary Materials.
The data for Figure 3 (A, B, C) should be added to the Supplementary Materials with at least the following information: UniProt Entry, protein name, EC, mol. mass, protein score based on proteomic analysis, gene name, GO.
Response 5- Yes, you are correct. Thank you for pointing that out. If Figure 3 assessed fold-change, then these additional values would indeed need to be considered. The term “fold change” used in the GO figures was incorrect. The correct terminology, according to Gene Ontology, is “fold enrichment,” and this has been updated in Figure 3. The fold enrichment value is calculated by GO from the list of gene names.
The data used to generate Figure 3 included all 253 SUMOylated peptides listed in Appendix Table A1. This table provides the gene name, the lysine with the GlyGly modification, and the PEP score, indicating detection and assignment confidence. No assessment of fold change was conducted for Figure 3. Gene names were used as identifiers because they are universally recognized across various platforms, including protein repositories, the protein data bank, STRING, and TheCellMap.
All proteomic data have been uploaded to the ProteomeXchange database.
Table 1 displays the changes in protein expression for SUMOylated proteins. Given the small dataset of 38 proteins, no fold change was selected. Additionally, we did not employ a q-value due to the small data set size. The decision to forgo typical cv and q-value cutoffs was made because the dataset is limited. In such, we wanted to report on proteins that were SUMOylated without change in response to simulated microgravity conditions. However, the p-values are included to highlight the power of SILAC labeling and the use of 6 biological replicates.
Comment 6- Q6. Page 9, Table 3. For statistical significance of protein fold change, cv (≤ 0.3) and q (≤ 0.05) values ​​should be assessed in addition to p (≤ 0.05) in the Supplementary materials. Add the UniProt Entry for each protein to Table 3.
Response 6- Table S2 has been added to show the UniProt entry, gene name, protein name, protein function, fold change, -log10 p-value, and q-value. Table S2 and Table 3 only show values that meet these thresholds: fc (≥ 0.58), cv (≤ 0.3), p (≤ 0.05), and q (≤ 0.05). This is also listed in Materials and Methods page 18, line 549.
Comment 7- Q7. Page 18, line 594: “saccharomyces cerevisiae” change on “Saccharomyces cerevisiae”
Response 7- Thank you for identifying this typo. It has been corrected.
Comment 8- Q8. Ln. 842. Provide page numbers for ref. 119
Response 8- Thank you for the attention. The reference is correctly listed for Int. J. Dev. Biol. 50: Effects of microgravity on cell cytoskeleton and embryogenesis Crawford-Young, S. J. (2006)
Round 2
Reviewer 1 Report
Comments and Suggestions for Authors
Revisions are accepted, and the manuscript is approved for publication.
Author Response
Thank you for your time and attention!
Reviewer 2 Report
Comments and Suggestions for Authors
I am not satisfied with authors response for my Q1 question.
Unfortunately, protein scores were not included either in the Material and Methods section or in the Table S2. A protein score in MS/MS is one of the main parameters indicating the level of confidence for the identified protein. All articles on proteomic analysis provide scores for each protein, allowing the reader to assess the confidence level of each individual protein. Authors must include the protein scores into the Table S2 for each protein.
Author Response
Comments- Unfortunately, protein scores were not included either in the Material and Methods section or in the Table S2. A protein score in MS/MS is one of the main parameters indicating the level of confidence for the identified protein. All articles on proteomic analysis provide scores for each protein, allowing the reader to assess the confidence level of each individual protein. Authors must include the protein scores into the Table S2 for each protein.
Response- Thank you for ensuring the data included in this manuscript was complete. In Table S2, the protein scores, peptide count, and percent sequence coverage has been added to the table.
Round 3
Reviewer 2 Report
Comments and Suggestions for Authors
Accept in the present form